# Towards End-to-End Image-to-Tree for Vasculature Modeling

**Manish Sharma**                                        SKMANISH@GOOGLE.COM
**Matthew C. H. Lee**                          MATTHEW.LEE13@IMPERIAL.AC.UK
**James Batten**                                    J.BATTEN@IMPERIAL.AC.UK
**Michiel Schaap**                                    MSCHAAP@HEARTFLOW.COM
**Ben Glocker**                                    B.GLOCKER@IMPERIAL.AC.UK

## Abstract

This work explores an end-to-end image-to-tree approach for extracting accurate representations of vessel structures which may be beneficial for diagnosis of stenosis (blockages) and modeling of blood flow. Current image segmentation approaches capture only an implicit representation, while this work utilizes a subscale U-Net to extract explicit tree representations from vascular scans. We obtain insights from these representations by associating tubular thickness with tree edges and visualizing the network of blood vessels from the Digital Retinal Vessel Extraction dataset (DRIVE).

## 1. Introduction

Imaging can be used to capture detailed information about complex anatomical structures such as blood vessel trees (also known as vasculature). This can help to detect conditions such as stenosis (Czarny and Resar, 2014), which is important for diagnosis and clinical decision making. Current approaches for extracting vasculature from images involve generating binary segmentation maps followed by further processing (Bates et al., 2017), (Chapman et al., 2015). However, these binary maps may be sub-optimal, implicit representations of the underlying geometry while trees seem a more natural way of describing vasculature. In this work, we propose a novel *image-to-tree* approach, which is an end-to-end system for extracting explicit tree representations of vasculature from biomedical scans. We designed a subscale U-Net based algorithm for predicting individual tree nodes and edges. The methodology is analysed on the Digital Retinal Vessel Extraction dataset (DRIVE). Using vascular tree construction, we discuss applications to thickness estimation in diabetic retinopathy prediction, and explore insights from visualizing these trees.

## 2. Image-to-tree for retinal images

Retinal vasculature contains crucial information about diseases like diabetes, macular degeneration, galucoma etc. For the purpose of this work, we consider the retinal scans from the Digital Retinal Vessel Extraction dataset, also called DRIVE. The dataset has a collection of manually annotated vessel segmentation maps for 20 images in the train and test set each. Additionally, we assume that a seed point on the optical disc is known. The algorithm works as follows - starting from an image patch around the seed point, a U-Net (Ronneberger et al., 2015) predicts the location of neighboring nodes, pushes them onto a

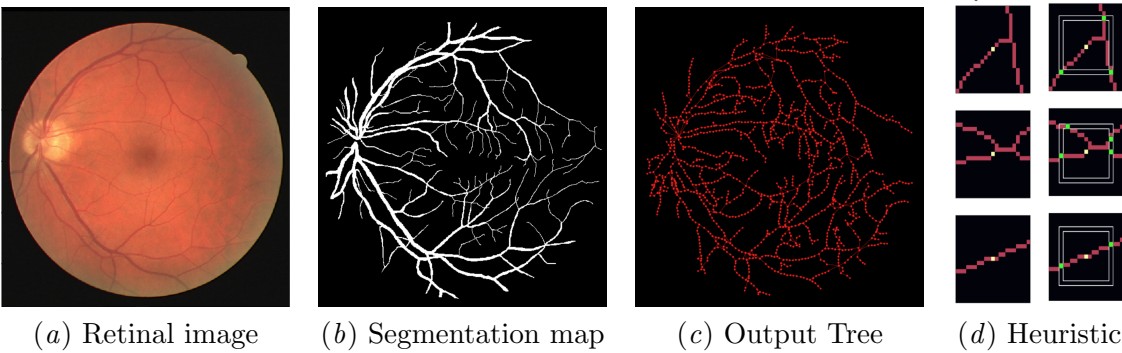

(*a*) Retinal image    (*b*) Segmentation map    (*c*) Output Tree    (*d*) Heuristic

Figure 1: (a, b, c) Input image, segmentation and output of Image-To-Tree for comparison. (d) Obtaining ground truth position of neighboring nodes to train subscale U-Net.

stack and incrementally moves the image patch anchor to the location at the top of the stack until it is empty. Predicted neighboring nodes are connected to get the tree structure.

In order to generate the data required to train our subscale U-Net (or the patch U-Net), we exploit the infomation from segmentation maps. First, we obtain a skeletonized version of the binary segmentation images (Lam et al., 1992), and use it to randomly sample many $21 \times 21$ size patches around valid points on the vessel network. Then, using a heuristic of minimum Chebyshev distance and traversal along the graph, we label the neighboring node locations as ground truth pixels (shown in Figure 1(*d*)). A U-Net is trained to identify these neighboring node locations from the corresponding patches in the RGB retinal image. The input of U-Net has shape $21 \times 21 \times 3$, and the output is a $21 \times 21$ probability map. At locations where the output map exceeds a certain threshold, we mark a neighbour node.

### 2.1. Evaluation

We do not have the notion of a "ground truth" tree, and hence use weak objective measures to determine the threshold for a pixel to be a neighboring node, and evaluate model's performance. Although not ideal, dice score with the skeletonized maps has a good correlation with the appropriateness of our tree. So we selected the threshold (and the model) with the highest dice score. At a sigmoid activation threshold of 0.3, we get an average recall of 0.62 and an average dice score of 0.4 with the skeletononized maps on the test set of 20 images in DRIVE database. Reconstruction output for a retinal image is shown in Figure 1(*c*). It must be noted that while the segmentation map is being utilized implicitly during the training process, inference is completely agnostic of it.

## 3. Applications and Visualization

Having obtained an explicit tree representation from a medical scan opens up new possibilities of vasculature analysis. In our work, we explored two such applications. First, we can analyze vessel thickness over the vasculature to obtain a model for blood flow, and highlight, for example, potential stenosis. We trained our subscale U-Net as a regressor to identify the thickness of predicted edges (along with neighboring nodes). Ground truth is provided

as the number of morphological thinning operations during the skeletonization phase. The U-Net was able to predict vessel thickness with a RMSE of 0.1 pixel, suggesting very good performance. Together with vessel type prediction (Kondermann et al., 2007), thickness prediction mechanism can be used to predict diabetic retinopathy which highly correlates to the arteriolar-to-venular width ratio (AVR). This can help detect eye blindness in early stages and assist clinical prognosis.

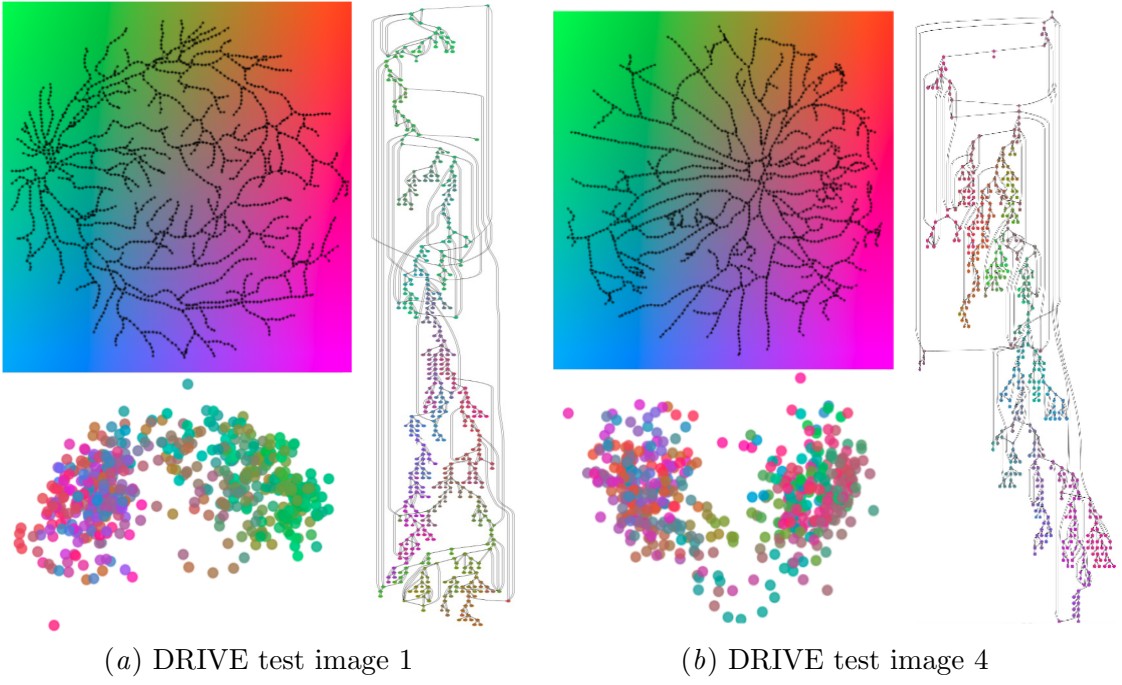

(a) DRIVE test image 1                    (b) DRIVE test image 4

Figure 2: Color coded nodes in original retinal image representation (left-top), GraphViz output (right) and node2vec output (left-bottom) for two images from DRIVE

Secondly, explicit tree representations can be studied from a geometrical and topological perspective. They can be morphed into other meaningful/interpretable spaces. Figure 2 shows the restructuring of tree nodes according to GraphViz (Ellson et al., 2001) and node2vec (Grover and Leskovec, 2016). Several notable insights can be drawn from these visualizations like: central retinal artery (or optic nerve) can be located at the intersection of semi-vasculatures obtained with node2vec; arteries and veins subsegments can be separated from the GraphViz visualizations.

## 4. Conclusion

We discussed a novel approach for extracting a tree from medical images. The algorithm uses a subscale U-Net to predict neighboring nodes from image patches. For retinal images, we discussed an application of vessel thickness prediction and obtained insights from visualizing vascular trees. Although the work is preliminary and accuracy of obtained trees are not as high as the state-of-the-art segmentation algorithms, we believe that extracing explicit geometric representations of vasculature is a promising direction for future research.

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
