# OpenReview forum: "Towards End-to-End Image-to-Tree for Vasculature Modeling"
_MIDL.io/2019/Conference/Abstract — MIDL Abstract 2019_

### Official Review · AnonReviewer1 · 2019-04-30
**Interesting idea, but concerns in method design, effectiveness and lack of baseline comparison**

**Rating:** 2
**Confidence:** 2

**Review:**

A U-net is used to incrementally predict vessel tree branch points in each image patch, effectively traversing all vessels in an image and thereby building a vessel tree. Methodology is questionable in case of parallel  and/or kissing vessels. It is evaluated on a rather outdated DRIVE dataset of color retinal images, while the evaluation is inconclusive as there is no baseline comparison.

Pros:
- The idea is interesting: going directly from images to trees instead of binary mask based segmentation with the use of neural networks to predict possible nodes in trees

Cons:
- Methodology does not seem to handle the case of parallel and/or kissing vessels, which may cause loops in the output vessel tree. This issues seems to have caused the cyclic loops evident from Figure 2.
- Evaluation is performed only on the DRIVE dataset, an old data set consisting of relatively small and old fundus images. The authors should consider the High-Resolution Fundus (HRF) image data set or the REVIEW data set.
- Author briefly mention an application of the proposed method for vessel thickness measurement, however, except for the RMSE score, no concrete expertimental setup and more insighful results are presented, nor is the RMSE result put into clinical context.

---

### Official Review · AnonReviewer2 · 2019-05-01
**Interesting novel application of a UNet for detecting  tree representations of vasculature**

**Rating:** 3
**Confidence:** 3

**Review:**

Interesting and novel idea to use a UNet to detect/predict individual tree nodes and edges instead of the common 2 step process of first segmenting the data and then compile it into a tree structure.

Starting from a known seed point, neighboring nodes are predicted from image patches centered at the current node. Detected nodes are pushed on the stack for further predictions. The input image patch is fed into a UNet with the output being a probability maps of node locations, which is thresholded to identify nodes.

The overall performance of the system is good (0.1 pixel RMSE)

While this is clearly early and somewhat limited work, the proposed approach is novel and shows promise

---

### Decision · Program_Chairs · 2019-05-06
**Acceptance Decision**

Accept